# Association between Age of Onset of Hypertension and Incident Atrial Fibrillation

**DOI:** 10.3390/jpm12071186

**Published:** 2022-07-21

**Authors:** Yonggu Lee, Jeong-Hun Shin, Byung Sik Kim, Hyungdon Kook, Woohyeun Kim, Ran Heo, Young-Hyo Lim, Jinho Shin, Chun Ki Kim, Jin-Kyu Park

**Affiliations:** 1Division of Cardiology, Department of Internal Medicine, Hanyang University Guri Hospital, Guri-si 11923, Korea; hmedi97@hanyang.ac.kr (Y.L.); hyapex@hanyang.ac.kr (J.-H.S.); fish3777@hanyang.ac.kr (B.S.K.); 2Division of Cardiology, Department of Internal Medicine, Hanyang University Medical Center, Seoul 04763, Korea; kookhyungdon@hanyang.ac.kr (H.K.); coincidence1@hanyang.ac.kr (W.K.); cardiohr@hanyang.ac.kr (R.H.); mdoim@hanyang.ac.kr (Y.-H.L.); jhs2003@hanyang.ac.kr (J.S.); 3Department of Medicine, Hanyang University College of Medicine, Seoul 04763, Korea; chunkikim@hanyang.ac.kr

**Keywords:** early-onset hypertension, atrial fibrillation, Ansan–Ansung cohort

## Abstract

We investigated whether age at hypertension (HTN) onset was associated with the risk of atrial fibrillation (AF) in the general population. This prospective longitudinal community-based cohort study included 9892 participants without AF at baseline, who underwent biennial electrocardiography for a median duration of 11.5 years. The participants were divided into five groups, consisting of a normotensive group (Group-N) and four HTN groups based on HTN onset age: <45 years (Group-H1); 45–54 years (Group-H2); 55–64 years (Group-H3); and ≥65 years (Group-H4). A multivariate Cox proportional hazards model showed that the presence of HTN at baseline was associated with higher AF risk (hazard ratio [HR], 1.93; 95% confidence interval [CI] 1.32–2.80). The participants in Group-H1 had the highest risk of AF (HR 3.18; CI 1.74–5.82), and the risk of AF decreased as HTN onset age increased across the four HTN groups (*p* for trend = 0.014). The AF onset age was significantly younger in participants in Group-H1 than in Groups-H2–H4. Early-onset HTN was associated with an increased risk of AF, and younger onset of AF in the general population. Surveillance for AF should be considered at a younger age in individuals with HTN.

## 1. Introduction

Atrial fibrillation (AF), the most common arrhythmia, increases the risk of stroke five-fold and the risk of cardiac failure three-fold, and eventually causes an increase in mortality rate [1]. Previously recognized predisposing factors for AF include aging, hypertension (HTN), obesity, very low or very high physical activity, and obstructive sleep apnea [2,3,4]. Among them, HTN is a major risk factor, contributing to one-fifth of newly diagnosed AF cases [5].

Cardiovascular diseases, the leading cause of death worldwide, accounted for 17.9 million deaths globally in 2019 [6]. HTN is one of the best-documented risk factors of cardiovascular diseases, along with diabetes [7]. HTN increases oxidative stress and stiffness of the arterial walls, and promotes endothelial dysfunction and inflammation, resulting in atherosclerotic changes in the arteries [8,9]. In both animal and human studies [10,11], these atherosclerotic changes induced fibrosis and remodeling of the left atrial wall, which promoted the subtle electrophysiological alterations that cause predisposition to AF. Recent large cohort studies have shown that a larger burden of high blood pressure (BP) and longer HTN duration are associated with increased risk of incident AF [12,13].

Exposure to HTN early in life may increase the risk of atherosclerotic cardiovascular diseases. Even in adolescence and young adulthood, early atherosclerotic changes, such as increased intima-media thickness, can occur in hypertensive patients [14]. Patients who develop HTN early in life reportedly face an increased risk of cardiovascular mortality compared to those who develop HTN later in life [15]. Suvila, et al., reported that early-onset HTN was associated with a higher prevalence of hypertensive end-organ damage, including left ventricular hypertrophy, diastolic dysfunction, coronary calcification, and albuminuria, and suggested the importance of assessing the age of HTN onset, to estimate the risk of hypertensive complications [16]. Given the high incidence of AF in patients with HTN, AF may be another manifestation of hypertensive end-organ damage, which may occur more frequently in patients who develop early-onset HTN. However, until now, no study has demonstrated a relationship between age at onset of HTN and the incidence of AF. Therefore, we investigated the association between the age of HTN onset and AF incidence in the general population, using the Korean Genome Epidemiology Study.

## 2. Materials and Methods

### 2.1. Study Population

This study was conducted on participants of the Ansan–Ansung cohort, which is a part of the Korean Genome Epidemiology Study, a large project funded by the Korean National Research Institute of Health, the Korean Centers for Disease Control and Prevention, and the Ministry of Health and Welfare, to investigate the genetic and environmental etiology of prevalent metabolic and cardiovascular diseases. The Ansan–Ansung cohort is part of an ongoing longitudinal cohort study being conducted on Koreans aged 40–69 years, first enrolled from June 2001 to January 2003, who reside in rural (Ansung) and urban (Ansan) communities. Detailed information regarding the study protocols has been reported in previous publications [3,4,17,18,19,20].

Participants with missing HTN onset data, who did not undergo electrocardiography (ECG) at baseline, and who were diagnosed with AF at baseline, were excluded from the study. Comprehensive health examinations, on-site interviews, and laboratory tests were conducted during the baseline visit, at a tertiary hospital located in the region. Six serial reassessments with the entire cohort protocol were performed through scheduled revisits every other year until 2014. All participants voluntarily enrolled in the study, and written informed consent was obtained from all participants. The study protocol adhered to the principles of the Declaration of Helsinki, and was approved by the Korean National Research Institute of Health and the Institutional Review Board (IRB No. HYUH 2017-12-033).

### 2.2. Assessment of Lifestyle and Medical History, Physical Examinations, and Laboratory Tests

A complete course of on-site interviews, that collected lifestyle and clinical information, and physical examinations were conducted by trained investigators at the tertiary hospital at every visit. Information on smoking, alcohol intake, education level, income level, marital status, and the presence of medical conditions including HTN, diabetes mellitus (DM), dyslipidemia, cerebrovascular disease, coronary artery disease, heart failure, and other previous diagnoses, was obtained using a questionnaire. Higher education was defined as a college degree or higher, and high income was defined as the median income of participants or higher. Physical activity and duration of physical activity were also assessed, using a questionnaire, and quantified using estimated daily metabolic equivalent task (MET) scores. AF prediction scores, including Cohorts for Heart and Aging Research in Genomic Epidemiology (CHARGE) AF score and Taiwan AF score, were calculated as previously reported [21,22].

Waist circumference (WC) was measured three times at the mid-level between the lowest rib and iliac crest at the end of expiration in a standing position, and averaged. BP was measured twice, using a mercury sphygmomanometer at the level of the heart in a sitting position, and was averaged. If the difference between the two BP measurements was ≥5 mmHg, another BP measurement was obtained, and the last two BP measurements were averaged.

Blood samples (10 mL) were collected from the antecubital vein after overnight fasting. Blood was collected and centrifuged at 1300× *g* for 10 min, and the serum was analyzed using an automated analyzer (Hitachi Automatic Analyzer 7600, Hitachi, Nittobo, Japan). Lipid profiles, hemoglobin A1c levels, and serum creatinine levels were measured. Participants were considered to have DM if they had been diagnosed with or were taking medications for DM or had a hemoglobin A1c level ≥ 6.5% [23]. Dyslipidemia was defined as: being diagnosed with dyslipidemia or taking statins without a history of cardiovascular disease or DM; having a total cholesterol level ≥ 240 mg/dL; having a triglyceride level ≥ 150 mg/dL; or having a high-density lipoprotein cholesterol level < 45 mg/dL.

### 2.3. Definition of HTN and Age of HTN Onset

A participant was considered to have HTN if they had been diagnosed with HTN by a physician. We did not include elevated BP levels, to restrict the definition of HTN to medically recognized HTN. The diagnosis of HTN and age of HTN onset were identified through recall, using the following questions in the questionnaire:For the diagnosis of HTN by physicians, “Have you ever been diagnosed with HTN by a physician?”For the age of HTN onset, “At what age were you first told by the physician that you had HTN?”

### 2.4. Electrocardiography and Identification of AF

Standard 12-lead electrocardiography (ECG) (GE Marquette MAC 5000^®^, GE Marquette Inc., Milwaukee, WI, USA) was performed in all participants at baseline and at every revisit. All ECG recordings were performed at a paper speed of 25 mm/s and an amplitude of 0.1 mV/mm, interpreted by a cardiologist, and coded according to the Minnesota code classification system.

AF was considered to be present when a participant presented with AF on ECG recordings, or if they had been diagnosed with AF by a physician before baseline or between visits. According to the Minnesota codes, the presence of AF was defined as 8-3-1, 8-3-2, 8-3-3, and 8-3-4. Newly developed AF was defined as the first identification of AF between visits, and the date of the new AF development was defined as the date when AF was first identified in ECG or the date when AF was diagnosed by a physician, if known.

### 2.5. Statistical Analysis

The participants were divided into a total of five groups, consisting of a normotensive group (Group-N) and 4 HTN groups based on the age of HTN onset: <45 years (Group-H1); 45–54 years (Group-H2); 55–64 years (Group-H3); and ≥65 years (Group-H4). Baseline characteristics of participants were compared among the groups, using one-way analysis of variance (ANOVA) for continuous variables—such as body mass index (BMI), WC, and low-density lipoprotein (LDL) cholesterol levels—and using Pearson’s chi-square test for categorical variables, such as sex, comorbidities, and smoking history. Post-hoc analyses of one-way ANOVA results were performed using Bonferroni correction. Continuous variables with a skewed distribution were compared between groups, using the Kruskal–Wallis test. Normality in the distribution of continuous variables was tested using the Shapiro–Wilk test.

The association between age of HTN onset and development of AF was evaluated using multivariate Cox proportional hazards models. Covariates for the multivariate Cox proportional hazards models included age, sex, residential area (rural vs. urban), household income ≥ median, BMI, central obesity (WC ≥ 90 cm for men and WC ≥ 85 cm for women), current smoking, current alcohol intake, physical activities (20.0–39.9 [reference level] vs. <20 vs. ≥40 MET-h/day), DM, myocardial infarction, heart failure, chronic lung disease, asthma, thyroid disease, and use of antihypertensive medications. Physical activity was categorized into three groups based on the restrictive cubic spline fit between physical activity as a continuous variable and AF risk. Multivariate Cox proportional hazards models were reduced through a backward variable selection procedure, and the cutoff criterion was set to *p* > 0.05 for individual variables to identify strong independent predictors of AF and reduce the overfitting bias in the models. Trends among the hazard ratios (HRs) of HTN onset age groups were tested using the Mann–Kendall trend test.

Age of AF onset was compared among all five groups (Groups-N–H4), and the interval between HTN onset and AF onset was compared among four HTN groups (Groups-H1–H4), using the Kruskal–Wallis test. Multivariate linear regression models were used to evaluate the association between the age of HTN onset and the age of AF onset in the presence of covariates.

Continuous variables were presented as mean ± SD, whereas variables with a skewed distribution were presented as median [interquartile range], including age (years), physical activity (MET-hours/day), duration of hypertension (years), triglyceride level (mg/dL), and C-reactive protein (mg/dL). Categorical variables were presented as numbers (%). All statistical analyses were performed using statistical software R-4.04 (R Core Team, R Foundation for Statistical Computing, Vienna, Austria) and its packages, including “survival”, “rms”, “tableone”, “trend”, and “descr”, in the RStudio-1.3 environment (RStudio Team, Rstudio, BPC, Boston, MA, USA). Statistical significance was set at *p* < 0.05.

## 3. Results

### 3.1. Baseline Characteristics of the Study Population

In total, 10,030 participants were enrolled during the baseline assessment period. Of these, 138 participants were excluded for one of the following reasons: missing age of HTN onset data (*n* = 64); no baseline ECG (*n* = 12); or baseline ECG showed AF (*n* = 62). Ultimately, 9892 participants were included in the analysis.

The mean age of the participants was 52.2 ± 8.9 years, and 5209 (52.7%) participants were female. The average BMI was 24.6 ± 3.1 kg/m^2^, and central obesity was found in 2919 (29.5%) participants. HTN was present in 1470 (14.9%) participants, of whom 1197 (81.4%) were taking antihypertensive medications. The median duration of HTN was 3.0 (interquartile range [IQR], 1.0–8.0) years. The age of HTN onset was <45 years in 351 participants (Group-H1), 45–54 years in 525 participants (Group-H2), 55–64 years in 479 participants (Group-H3), and ≥65 years in 115 participants (Group-H4).

The baseline characteristics of the participants divided into Groups-N–H4 are described in Table 1. Most baseline characteristics were significantly different among the five groups, except for physical activity and the presence of thyroid disease, asthma, and chronic lung disease. Compared to Group-N, the participants in Group-H1 were more likely to be male and urban residents, and to have had higher education and higher income, whereas the participants in Groups-H2–H4 were more likely to be female and rural residents, and to have had lower education and lower income. Compared to Groups-H2–H4, a higher percentage of participants in Group-H1 were current smokers as well as current drinkers, whereas a lower percentage of participants in Group-H1 were taking antihypertensive medications. Compared to the participants in Group-N, participants in each of the four hypertensive groups (Groups-H1–H4) had higher BMI and were more likely to have central obesity, DM, dyslipidemia, myocardial infarction, and cerebrovascular disease. However, the frequency of comorbidities did not differ among Groups-H1–H4. The CHARGE AF score and the Taiwan AF score were lowest in Group-N, and gradually increased as the age of HTN onset increased. In hypertensive participants, both risk scores for AF were lowest in Group-H1, and were similar among Groups-H2–H4. In general, serum fasting glucose, hemoglobin A1c, total cholesterol, LDL cholesterol, and C-reactive protein levels were higher in Groups-H1–H4 than in Group-N. However, there were no significant differences in laboratory data among Groups-H1–H4, except that estimated glomerular filtration rate levels were higher in Group-H1.

### 3.2. The Risk of AF According to Age at HTN Onset

Over a median follow-up of 11.5 years (138 months, IQR 68–141 months), new-onset AF occurred in 148 participants; the incidence was 1.714 per 1000 person-years (148 AF cases/86,304.4 person-years). AF incidence was significantly higher in the combined hypertensive Group-H (39/1470, 2.7%) than in Group-N (109/8422, 1.3%). Among the four separate hypertensive groups, the highest incidence was observed in Group-H1 (3.4%), and the incidence of AF in Groups-H2–H4 ranged from 2.3% to 2.6% (Figure 1). In univariate Cox proportional models, the risk of incident AF presented no trend among Groups-H1–H4 (*p* for trend = 0.327). Univariate Cox proportional hazards models for the relationship between covariates and the risk of incident AF are described in Appendix A. The risk of incident AF increased almost linearly with age, had a V-shaped biphasic relationship with physical activity, and was significantly higher in men than in women. None of the other covariates were significantly associated with the risk of incident AF. In multivariate models, the risk of incident AF in the combined hypertensive group (Groups-H1–H4) remained higher than in Group-N.

However, when each hypertensive group was analyzed separately, the risk was highest in Group-H1, and gradually decreased to the lowest in Group-H4 (*p* for trend = 0.014). With Group-N as the reference group, the risk of incident AF was significantly higher in Group-H1, while that in Group-H2 approached statistical significance. The risk of incident AF in Group-H3 or Group-H4 was not significantly higher than in Group-N (Figure 1). Age, male sex, and physical activity <20.0 or ≥40 MET-h/day were significantly associated with increased risk of incident AF in the multivariate model (Appendix A).

### 3.3. Age of AF Onset According to the Age of HTN Onset

The median age of AF onset was 63 years [IQR, 54–70 years] in all 148 participants with new-onset AF. The age of AF onset in Group-H1 was significantly lower than in Group-N: (51 years [IQR, 48–60 years] vs. 62 years [IQR, 53–70 years]; *p* = 0.004). The age of AF onset in Group-H3 was higher than in Group-N: (68 years [IQR, 65–73 years] vs. 62 years [IQR, 53–70 years]; *p* = 0.015). Among hypertensive participants, the age of AF onset in Group-H1 was significantly lower than in the other hypertensive groups (*p* = 0.002, vs. Group-H2; *p* < 0.001 vs. Group-H3; *p* = 0.020 vs. Group-H4), and there were no differences in the age of AF onset among Groups-H2-4. The age of AF onset was not significantly different between Group-H2 and Group-N, nor between Group-H4 and Group-N (Figure 2). In age- and sex-matched analysis, the significant difference in age of AF onset between Group-N and Group-H1 was maintained (Appendix A). The proportion of participants diagnosed with AF at <65 years was higher in Group-H1 than in Group-N (91.7% vs. 58.7%; *p* = 0.026) (Figure 2). A linear regression model showed that an age of HTN onset <45 years was associated with a 9.66-year (95% CI, 4.39–14.95) younger age of AF onset in participants developing new-onset AF (Table 2). A multivariate model also showed that an age of HTN onset < 45 years remained significantly associated with a decreased age of AF onset by 1.79 (95% CI, 0.03–3.55) years, after adjusting for age at baseline and other covariates.

### 3.4. Intervals between HTN Onset and AF Onset

Median intervals between HTN onset and AF onset in individual hypertensive groups ranged from 5 years to 14.5 years (Figure 3). Comparison of the intervals among groups may not be meaningful, given the small sample size in each group. Nonetheless, the interval between HTN onset and AF onset in Group-H1 was not significantly different from that in Groups-H2–H4 individually, nor from that in Groups-H2–H4 combined. If anything, it was shorter than in Group-H2, although statistically not significant, possibly due to the small sample size (8.5 years [IQR, 7.0–14.5 years] vs. 14.5 years [IQR, 10–21.3 years]; *p* = 0.099).

## 4. Discussion

We found that HTN was associated with an approximately two-fold increased risk of AF, and that an age of HTN onset < 45 years was associated with a greater increase in risk of AF than an older age of HTN onset. In participants with an age of HTN onset < 45 years, the interval between HTN onset and AF onset was 8.5 years, which was not significantly longer than in other hypertensive participants with an age of HTN onset ≥ 45 years.

An association between AF and HTN has been demonstrated in many epidemiological studies. A recent study in the Atherosclerosis Risk in Communities cohort, using Mendelian randomization causal inference, showed that the incidence of AF was higher by 50% in patients with HTN [24]. Lee, et al., showed that a larger burden of high BP was associated with a higher incidence of AF in the general population [12]. In a large population-based cohort study, Kim, et al., reported that the risk of AF increased in patients with a longer duration of HTN [13]. A longer duration of HTN in similarly aged patients translates to early onset of HTN; however, to date, a direct association between age of HTN onset and risk of incident AF has not been reported. This study is the first to show that early-onset HTN is associated with higher risk of AF and younger onset of AF.

Younger HTN onset has been reported to affect a patient’s health more severely than HTN starting in middle age or later [15]. Suvila, et al., reported that patients with HTN onset at age < 35 years had an odds ratios of 2.29 for left ventricular hypertrophy, 2.06 for diastolic dysfunction, and 2.94 for coronary calcification, respectively, compared to normotensive individuals, whereas HTN onset at age > 45 years was not related to increases in target organ disease after 15 years of follow-up [16]. Kishi, et al., also reported that cumulative exposure to high BP over long periods from young adulthood was associated with increased risk of left ventricular systolic and diastolic dysfunction in middle age [25]. Left atrial cardiomyopathy due to structural and functional changes in the left ventricle has a pathophysiological association with AF. An animal study suggested that HTN promotes left ventricular remodeling, which is accompanied by atrial cardiomyopathy, indicating a potential mechanism underlying this association [10]. The duration of exposure to high BP has recently been suggested to be a risk factor mediating the relationship between HTN onset age and incident AF. HTN occurring at a younger age would provide more time for myocardium to develop remodeling processes by increased afterload and arterial stiffness, while it would also increase time for arterial walls to develop more stiffness. Our study also showed that the risk of incident AF among hypertensive participants decreased with increasing age at HTN onset.

The strong impact of aging on AF development has been reported in numerous studies. Our results also demonstrated the strong association between age and incident AF. After adjusting for covariates, including age at baseline, the HR for incident AF among hypertensive participants decreased most remarkably in those with an HTN onset ≥ 65 years in our study. Linear regression analysis also showed a remarkable decrease in the absolute value of the coefficient of an age of HTN onset < 45 years after adjusting for age, although an age of HTN onset < 45 years remained a significant determinant of AF onset age.

According to data extracted from the Korea National Health and Nutrition Examination Survey (a nationwide representative population-based survey), the prevalence of HTN in the youngest group (under 30 years of age) was 8.6%, which is lower than those in the other age groups [26]. Also, the rates of awareness, treatment, and control were poor in the youngest group. In our study, participants with early HTN onset were younger than those with late HTN onset. They had the lowest rate of control with antihypertensive medications. Additionally, participants with early HTN onset had unhealthy lifestyles, such as frequent alcohol intake and smoking. These findings imply that young patients with HTN may tend to overlook adverse outcomes developing later in life. Our study suggests that HTN onset age provides valuable information in determining the risk of incident AF in the management of HTN patients. However, to date, there has been no recommendation by current guidelines that incorporated the age of HTN onset as part of risk assessment [2,27]. In the 2020 AF guideline [2], opportunistic AF screening was recommended for the elderly population (≥65 years). Given that early HTN onset carries a stronger risk of incident AF, and that most AF develops before the age of 65 years, earlier screening is warranted in patients who develop HTN at a younger age.

Our results showed that among hypertensive participants, the CHARGE AF score and the Taiwan AF score for predicting incident AF were lowest in Group-H1 and highest in Group-H4, whereas the actual incidence of AF was highest in Group-H1. This finding may indicate the potential usefulness of age at HTN onset as a predictor of AF, to improve the performance of these AF prediction models, because the age of HTN onset could reflect the young patients at risk of AF underrepresented by these AF prediction models.

The strengths of our study include its large sample size and long-term follow-up. In addition, the quality of this study was enhanced by conducting face-to-face interviews and BP measurements at each examination, in strict observance of the standardized protocol biennially. We identified AF using a self-reported history of physician-determined diagnosis, as well as biennial ECG. This approach enabled reasonably sensitive identification of AF, especially in asymptomatic patients. Therefore, we could analyze the time interval between the onset of HTN and AF in greater detail.

### Limitations

This study had several limitations. Firstly, our study was observational; therefore, cautious interpretation of causal relationships is needed. Although we adjusted for relevant covariates—including age, sex, socioeconomic status, and comorbidities—unknown confounding factors may have been partly responsible for the association between early-onset HTN and incident AF. Secondly, information regarding the age of HTN onset was obtained via self-reporting. Therefore, recall bias may have been present. However, the use of self-reporting to determine the age of HTN onset has been shown to be reasonably reliable by Suvila, et al. [28]. Thirdly, the incidence of AF (1.714/1000 person-years) was quite low compared with other cohort studies [29,30], which may indicate under-recognition of AF. In particular, because we investigated incident AF through 12-lead standard ECG obtained biennially, and self-reported physician diagnosis, some paroxysmal AF may not have been recognized. However, Northeast Asians have the lowest prevalence of AF in the world [31], and a previous study by Lee, et al., reported a similarly low incidence of AF at 1.5–1.7/1000 person-years, from a nationwide insurance claims database in the Korean population [32]. Therefore, under-recognition of paroxysmal AF may not have had a significant impact on the findings of this study. Finally, the age of HTN onset and AF onset do not necessarily represent the actual timing of HTN or AF development, but rather when these conditions were first diagnosed by physicians (age of HTN onset) or cohort investigators (incident AF). Presumably, the difference between the actual onset and diagnosis was evenly distributed across the cohort population, but may have been skewed by changes in participant behavior after HTN diagnosis.

## 5. Conclusions

This prospective cohort study provides additional compelling evidence regarding the adverse effects of early HTN onset on incident AF. Early HTN onset is a strong risk factor for incident AF; by contrast, this relationship was gradually attenuated with the increasing age of HTN onset. Compared with the normotensive group, those with early-onset HTN had an onset of AF at a significantly younger age, with a median (IQR) of 51 (48–71) years, which suggests that subjects with early-onset HTN may be potential candidates for more meticulous screening of AF. Further prospective research is warranted to clarify whether early intervention, including modification of unhealthy lifestyles and antihypertensive medication for patients with early HTN onset, is more beneficial for preventing AF.

## Figures and Tables

**Figure 1 jpm-12-01186-f001:**
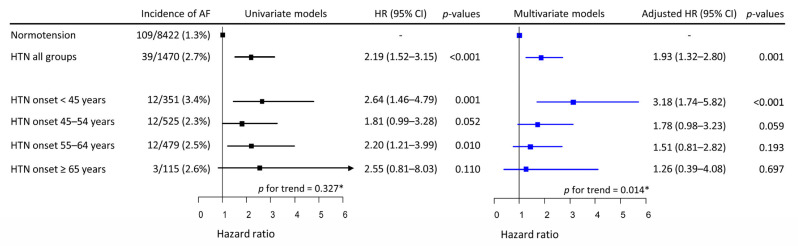
Univariate and multivariate Cox proportional hazards models of HTN onset age for incident AF. HTN was associated with increased risk of incident AF by approximately a factor of 2. In the univariate model, HTN onset at any age appeared to be associated with a higher risk of incident AF, while the risk of AF did not vary with the age of HTN onset. However, in the multivariate model, the risk of incident AF increased inversely with the age of HTN onset. * Mann–Kendell test. HTN, hypertension; HR, hazard ratio; CI, confidence interval; AF, atrial fibrillation.

**Figure 2 jpm-12-01186-f002:**
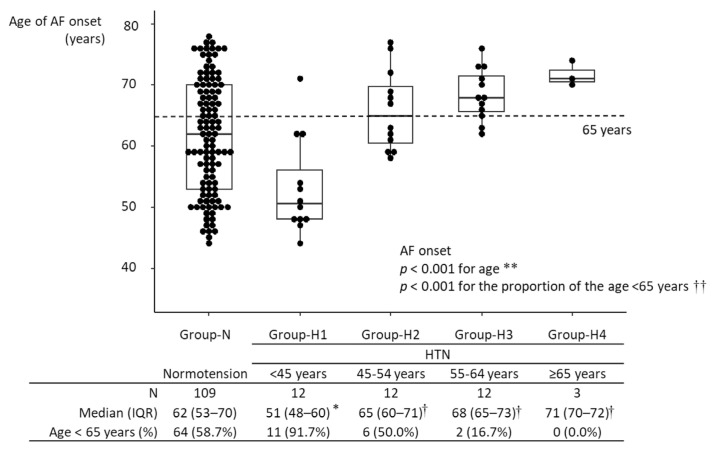
Age of AF onset in the different groups. * *p* < 0.05 vs. Group-N; † *p* < 0.05 vs. Group-H1. During the follow-up period, 148 participants developed AF. The age of AF onset ranged from 44 to 78 years, with a median of 63 years (IQR: 54–70 years). Among hypertensive participants, the age of AF onset was lowest in Group-H1 and highest in Group-H4. The age of AF onset in Group-N significantly differed only from that in Group-H1. Among hypertensive participants, the age of AF onset in Group-H1 was significantly lower than those in the other groups, and there were no significant differences in the age of AF onset among Group-H2-4. In Group-H1, the age of AF onset was <65 years in >90% of participants. ** Kruskal–Wallis test; †† Fisher’s exact test. HTN, hypertension; IQR, interquartile range; AF, atrial fibrillation.

**Figure 3 jpm-12-01186-f003:**
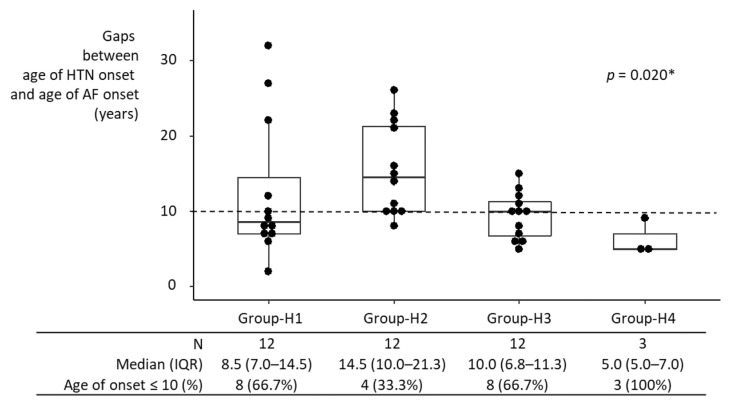
Interval between HTN onset and AF onset in hypertensive participants. AF developed in 39 of 1470 participants who were diagnosed with HTN at baseline. The interval between HTN onset and AF onset in Group-H1 was not significantly different from Groups-H2–H4 individually, nor from that in Groups-H2–H4 combined. If anything, it was shorter than in Group-H2, although this was not statistically significant (*p* = 0.099), possibly due to the small sample size. HTN, hypertension; IQR, interquartile range; AF, atrial fibrillation; * Kruskal–Wallis test.

**Table 1 jpm-12-01186-t001:** Baseline characteristics of participants.

	Age of HTN Onset					
	Normotension	<45 Years	45–54 Years	55–64 Years	≥65 Years	*p*-Values
	*N* = 8422	*N* = 351	*N* = 525	*N* = 479	*N* = 115	
	(Group-N)	(Group-H1)	(Group-H2)	(Group-H3)	(Group-H4)	
Age (years)	49 [44, 59]	46 [43, 51]	54 [51, 59]	63 [61, 66]	68 [67, 69]	<0.001
Female sex	4349 (51.6)	152 (43.3)	321 (61.1)	318 (66.4)	69 (60)	<0.001
Urban residential area	4375 (51.9)	209 (59.5)	213 (40.6)	142 (29.6)	31 (27)	<0.001
High education	1120 (13.3)	94 (26.8)	51 (9.7)	32 (6.7)	6 (5.2)	<0.001
Income ≥Median	4347 (51.6)	224 (63.8)	212 (40.4)	110 (23)	16 (13.9)	<0.001
BMI (kg/m^2^)	24.3 ± 3.1	26.1 ± 3.2	26.3 ± 3.1	25.8 ± 3.2	25.4 ± 2.9	<0.001
WC (cm)	81.9 ± 8.7	85.9 ± 8.4	87.4 ± 8.1	87.8 ± 8.3	87.2 ± 8.4	<0.001
Central obesity	2179 (25.9)	151 (43.0)	267 (50.9)	261 (54.5)	61 (53.0)	<0.001
Smoking						<0.001
Never a smoker	4882 (58.0)	180 (51.3)	357 (68.0)	324 (67.6)	79 (68.7)	
Ex-smoker	1270 (15.1)	72 (20.5)	97 (18.5)	78 (16.3)	19 (16.5)	
Current smoker	2270 (27.0)	99 (28.2)	71 (13.5)	77 (16.1)	17 (14.8)	
Alcohol intake						<0.001
Never a drinker	3837 (45.6)	125 (35.6)	269 (51.2)	285 (59.5)	68 (59.1)	
Ex-drinker	525 (6.2)	24 (6.8)	42 (8.0)	53 (11.1)	7 (6.1)	
Current drinker	4060 (48.2)	202 (57.5)	214 (40.8)	141 (29.4)	40 (34.8)	
PA (MET-hr/day)	19.0 [11.3, 35.0]	18.0 [11.3, 26.1]	18.8 [11.3, 35.4]	19.1 [11.1, 35.8]	20.3 [9.0, 38.8]	0.431
Systolic BP (mmHg)	121.6 ± 16.9	139.7 ± 20.0	141.9 ± 18.4	144.7 ± 18.8	144.0 ± 18.1	<0.001
Diastolic BP (mmHg)	81.2 ± 11.0	92.8 ± 11.8	92.6 ± 11.4	90.1 ± 10.7	89.2 ± 10.6	<0.001
Antihypertensive drugs	92 (1.10)	235 (67.0)	397 (75.6)	383 (80.0)	90 (78.3)	<0.001
Duration of HTN (years)		6.0 [2.0, 14.0]	4.0 [1.0, 9.0]	3.0 [1.0, 6.0]	1.0 [0.5, 3.0]	<0.001
Comorbidity						
DM	785 (9.3)	83 (23.6)	112 (21.3)	135 (28.2)	31 (27)	<0.001
Dyslipidemia	5695 (67.6)	286 (81.5)	419 (79.8)	385 (80.4)	91 (79.1)	<0.001
Myocardial infarction	50 (0.6)	6 (1.7)	15 (2.9)	14 (2.9)	1 (0.9)	<0.001
Heart failure	12 (0.1)	0 (0.0)	3 (0.6)	4 (0.8)	1 (0.9)	0.001
Coronary artery disease	48 (0.6)	4 (1.1)	9 (1.7)	15 (3.1)	1 (0.9)	<0.001
Stroke	62 (0.7)	8 (2.3)	21 (4.0)	20 (4.2)	4 (3.5)	<0.001
Thyroid disease	244 (2.9)	12 (3.4)	23 (4.4)	19 (4.0)	3 (2.6)	0.247
Asthma	177 (2.1)	7 (2.0)	17 (3.2)	12 (2.5)	3 (2.6)	0.495
Chronic lung disease	49 (0.6)	2 (0.6)	7 (1.3)	5 (1.0)	2 (1.7)	0.107
AF risk score						
CHARGE AF score (ΣXβ)	10.0 [9.4, 10.9]	10.2 [9.6, 10.8]	10.8 [10.3, 11.4]	11.7 [11.3, 12.1]	12.1 [11.8, 12.4]	<0.001
CHARGE AF 5-year risk (%)	0.22 [0.12, 0.53]	0.25 [0.15, 0.46]	0.49 [0.31, 0.83]	1.16 [0.79, 1.69]	1.68 [1.34, 2.23]	<0.001
Taiwan AF score	1 [−1, 2]	1 [0, 2]	2 [1, 3]	4 [3, 4]	4 [4, 6]	<0.001
Laboratory data						
eGFR (mL/min/1.73 m^2^)	93.1 ± 13.8	90.3 ± 17.8	88.2 ± 15.7	83.8 ± 14.7	78.8 ± 14.3	<0.001
FBS (mg/dL)	87.5 ± 22.2	96.6 ± 29.6	93.8 ± 26.7	94.1 ± 26.3	92.1 ± 25.1	<0.001
PC2h glucose (mg/dL)	127.3 ± 56.7	150.8 ± 72.0	150.3 ± 67.8	159.6 ± 69.0	153.6 ± 71.6	<0.001
HbA1c (%)	5.7 ± 0.9	6.0 ± 1.2	6.0 ± 1.1	6.2 ± 1.1	6.1 ± 1.0	<0.001
Total-C (mg/dL)	190.0 ± 35.7	201.3 ± 36.7	198.2 ± 36.4	195.5 ± 35.4	200.4 ± 40.3	<0.001
HDL-C (mg/dL)	45.0 ± 10.1	43.2 ± 9.4	42.7 ± 9.6	43.1 ± 10.0	44.6 ± 11.5	<0.001
Triglyceride (mg/dL)	131 [97, 1840]	159 [113, 224]	161 [117, 225]	160 [124, 218]	159 [120, 228]	<0.001
LDL-C (mg/dL)	118.0 ± 30.5	127.5 ± 31.3	124.8 ± 31.8	122.4 ± 30.8	124.8 ± 33.7	<0.001
CRP (mg/dL)	0.14 [0.06, 0.24]	0.16 [0.08, 0.28]	0.18 [0.09, 0.31]	0.18 [0.10, 0.32]	0.18 [0.09, 0.36]	<0.001

Data are presented as the mean ± SD or number (%). For variables with a skewed distribution, the data are presented as the median [1st quartile value, 3rd quartile value]. HTN, hypertension; MET, metabolic equivalent task; BP, blood pressure; DM, diabetes mellitus; eGFR, estimated glomerular filtration rate; HDL-C, high-density lipoprotein cholesterol; LDL-C, low-density lipoprotein cholesterol; CRP, C-reactive protein; BMI, body mass index; WC, waist circumference; FBS, fasting blood sugar.

**Table 2 jpm-12-01186-t002:** Determinants of the Age of AF Onset in Hypertensive Participants.

	Model Summary	Determinants	Coefficient	SE	*p*-Values
Univariate	Adjusted *R*^2^ = 0.074	Age of HTN onset < 45 years	−9.66	2.70	0.0005
Multivariate	Adjusted *R*^2^ = 0.908	Age of HTN onset < 45 years	−1.79	0.90	0.0473
		Age (years)	0.97	0.03	<0.0001
		History of Asthma	−3.06	1.47	0.0381

Covariates included age, sex, residential area, household income, BMI, central obesity, current smoking, current alcohol intake, physical activity, history of DM, myocardial infarction, heart failure, asthma, chronic lung disease and thyroid disease, and use of anti-hypertensive medications. The multivariate model was reduced through a backward variable selection procedure (cutoff criteria *p* > 0.05). HTN, hypertension; DM, diabetes mellitus; BMI, body mass index.

## Data Availability

Access to the data is regulated by the Korean Centers for Disease Control and Prevention. The data are open to any researchers without charge, if the study protocol is approved by appropriate ethics committees, and applications for research permission can be filed on the website of the Korean Centers for Disease Control and Prevention (http://nih.go.kr/contents.es?mid=a40504060100; accessed on 20 July 2022). The unedited complete R scripts used for the statistical analyses were uploaded to the following data repository (https://osf.io/dczpf; accessed on 20 July 2022). The R scripts can also be requested from the author, Yonggu Lee, M.D. (hmedi97@hanyang.ac.kr).

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
