# Peer review of "Association between Age of Onset of Hypertension and Incident Atrial Fibrillation"

_jpm, 2022, doi:10.3390/jpm12071186_

Round 1

Reviewer 1 Report

Lee et al. submitted a quite interesting original research article describing the association between the age of arterial hypertension onset and incidence of atrial fibrillation. This is an important subject because cardiovascular diseases are one of the most important problems for healthcare systems worldwide. The attractiveness of the work is additionally increased by factors such, as: a relatively large group of study participants, a relatively long observation period, the use of relatively rarely collected data for the analysis, such as the age at which hypertension occurred and the age at which atrial fibrillation occurred. Although the article is quite well prepared, in my opinion, some modifications are needed to further improve the quality of the manuscript.

Major recommendations:

1) The introduction is too short and laconic. In my opinion, some general information about cardiovascular diseases (also in the course of atherosclerosis) should be presented to elucidate how important it is for contemporary medical practice. Then, information about arterial hypertension should be mentioned. Arterial hypertension is a risk factor for the development of atherosclerosis, which may lead to cerebrovascular diseases, including ischemic stroke, for which atrial fibrillation is also a significant risk factor. It shows how strictly different pathological processes are involved in the development of cardiovascular diseases. See, for example, the following references taken from the high-quality and up-to-date scientific literature: doi.org/10.3390/ijerph182211970; doi.org/10.3390/ijerph17249339; doi.org/10.1155/2018/1098039; doi.org/10.1155/2020/6471098; doi.org/10.3390/ijerph13020201; doi.org/10.3390/antiox11010172; doi.org/10.3390/biom11040585.

2) I believe that too few references were cited in the work. The 21 cited works, for such a widely researched and discussed subject, put into question the insight of the discussion.

3) In the section on the methodology of statistical analysis, please describe how the compliance of the distribution of a given variable with the normal distribution was examined.

4) “Data are presented as mean ± standard deviation or numbers (%).” (line 144) It should be clearly demonstrated for which variables the distribution is consistent with the normal distribution and for which it is not. The mean and standard deviation should be used for variables that are normally distributed, and the median and the interquartile range should be used for variables whose distribution is not normal.

5) All limitations of the study should be discussed.

 The list of references should be prepared in accordance with the rules of MDPI.

English quality should be revised by a specialist.

Reviewer 2 Report

Dear Sir/Madam,

 I had the opportunity to act as a reviewer on the recent submission by Lee et al. to the Journal of Personalized Medicine.

The authors present interesting research on the epidemiology and especially incidence of atrial fibrillation in patients already diagnosed with systemic hypertension. They showed that early-onset hypertension was associated with an increased risk of atrial fibrillation and onset of atrial fibrillation.

The manuscript is very well written and the results are interesting and of high clinical interest.

However, I recommend addressing following (minor) issues:

  1. How do the scores for incident atrial fibrillation (e.g., Framingham, CHARGE AF or TAIWAN AF) perform in the cohort?
  2. What could be the possible explanation for the BMI curve shape in Figure S1?
  3. Regarding the definition of diabetes with HbA1c (line 90): please provide reference.
  4. How is the incidence on line 181 compared with the known values in the literature: higher or lower?

 Best regards,

Round 2

Reviewer 1 Report

I received for review a revised version of the manuscript prepared by Lee et al. entitled: "Association Between Age of Onset of Hypertension and Incident Atrial Fibrillation".

In my opinion, P. T. The Authors of the publication responded well to the comments contained in the review and improved the manuscript in a satisfactory manner. I recommend the paper for publication in its current form.

Thank you for inviting me to the review. I congratulate the authors and wish them success in their further scientific work.